# Regional Population and Public Services under the Framework of Sustainable Development: Evidence from a Typical High-Tech Zone in China

**Xueyi Wang, Taiyi He \* and Ke Li**

Research Institute of Social Development, Southwestern University of Finance and Economics, Chengdu 610074, China; wangxy2616@sina.com (X.W.); keli918@163.com (K.L.)
\* Correspondence: hty13628092628@126.com

**Abstract:** As organic parts of regional systems, the development and connection of the population and public services are of great significance to the realization of regional sustainable development. As the typical development sample of regional systems in China, the development and connection of the population and public services in high-tech zones deserve more attention. This paper takes the population and public services of the typical high-tech zone in western China as its research object, and uses the entropy method and the coupling coordination degree model to measure the development level, correlation degree and effect of population and public services in the high-tech zone from 2012 to 2021. The empirical results show that: firstly, the development level of the population system in the high-tech zone shows a positive evolutionary trend in the sample period, and the population system development makes a stable contribution to the sustainable development in the high-tech zone. Secondly, the development level of the public services system in the high-tech zone shows a dynamic evolutionary trend in the sample period. The contribution of the public services system to the sustainable development in the high-tech zone is affected by the population system. Thirdly, the correlation degree between the population and public services systems in the high-tech zone is kept at a relatively high level during the sample period, and the two subsystems have a relatively close element fusion. At the same time, the relationship between the population and public services systems is optimized in the sample period. There is positive information exchange, valuable feedback and dynamic adjustment between the two subsystems. The research implication is to dialectically grasp the development and connection between regional population and public services systems under the framework of sustainable development.

**Keywords:** sustainable development; system; population; public services; high-tech zone

## 1. Introduction

Sustainable development is a global reform that takes the population, economy, society and environment into account [1]. At the elementary stage, the definition of sustainable development has generated intense debate, and research has led to a consensus that sustainable development is a model for preserving contemporary values without compromising future values [2]. With the advancement in social development, sustainable development has been brought into a broader discussion framework. For example, as an integral part of the Millennium Development Goals (MDG), sustainable development aims to promote poverty reduction, education and global partnership for development in global systems engineering [3]. Since entering the new era, sustainable development has brought a broader opportunity, that is, guiding multi-elements under the unified criteria of positive development [4].

The value of sustainable development in guiding regional development is becoming more and more prominent, which leads to a new issue, that is, how to realize sustainable development [5]. Systems theory can provide a new angle for understanding sustainable

development [6]. Systems theory follows the basic principle of systems science, which has the main characteristics of integrity, emergence, openness and causality. From the perspective of representation, there are many overlaps between systems theory and sustainable development, so it has a value in being introduced [7]. From the perspective of systems theory, sustainable development is a non-static and non-one-sided way of looking at things, and it is subject to multiple relationships and constraints that help to clarify the difficulty of achieving goals and to integrate synergies among the goals, antagonisms and coherence between measures and goals [8,9].

From the perspective of systems theory, the inevitable requirement of realizing regional sustainable development is coordination between regional systems. From the perspective of previous research, coordination among the economic, resource and environmental systems has been proved to be important [10,11]. In the process of reviewing sustainable development, it is found that Article 16 clearly stated that public management engineering is very important in sustainable development [12]. At present, the representative public management projects are population and public services, and the optimization of population and public services systems will accelerate the sustainable development of public management projects [13]. By contrast, population and public services systems that have value in this subject have not appeared frequently in the discussion of the existing literature.

The population system is a general term of value to analyze the object of population, and can be divided into internal and external systems [14]. The sustainability of the population system is usually manifested in the optimization of the spatial distribution of the population, the improvement in the educational level of the population and the enhancement of the well-being of the population [15–18]. At the same time, the public services system is a general term of value to analyze public goods and services which the public organization provides and has a common consumption nature represented in its general name—it covers goods, services, organizations, etc. [19,20]. The sustainability of the public services system is reflected in the increase in educational resources [21], the optimization of traffic conditions [22,23], the health care system [24], the cultural level of the area [25], the consolidation of social security functions [26], and the improvement in digital services capacity [27]. The contribution of population and public services systems in regional systems shows that the former provides dynamic support for the sustainable development of regional systems, while the latter provides a process guarantee for the sustainable development of regional systems [28].

The value of population and public services systems lies not only in their independence but also in their connection [29]. In China, for example, the early public services in China were linked to the household registration system. With the expansion and acceleration of population migration, the supply of public services is facing a disconnection between scale and quality, and the rational allocation of regional resources and sustainable development is facing great challenges [30]. Therefore, the effect and intensity of population and public services are of great value for the achievement of sustainable development.

There is a large development gap between regions in China [31], so it is difficult to explain regional developments with different characteristics in a theoretical framework. Selecting a representative region is the way forward. This research chooses a high-tech representative zone located in western China, namely Chengdu, as the research object. First, the high-tech zone of Chengdu has the main characteristics of other high-tech zones, such as a large population, a large floating population, a high degree of industrial agglomeration, relatively large scientific and technological talents, a high population quality, a relatively young population structure, strong support for digital innovation, and a relatively clear policy bias [32–36]. Secondly, the high-tech zone of Chengdu is the driving force of the development of science and technology in the Western Region of China, and plays an exemplary role in the process of building regional economic growth, having a knock-on effect on the external economy [37]; for example, through reducing the degree of resource mismatch and improving total factor production [38,39]. Compared with other high-tech zones, the irreplaceable feature of the high-tech zone of Chengdu is more significant.

Based on the literature review, the two problems deserve to be discussed in depth. Firstly, what is the level of population and public service development in a typical high-tech zone in western China? Secondly, what is the relationship between the population and public service development in a typical high-tech zone in western China? In order to reply to this problem, this paper aims to analyze the population and public services system of a typical high-tech zone in western China under the framework of sustainable development.

The novelty of this paper lies in: firstly, the systems theory is introduced as the cognitive perspective of sustainable development, which broadens the scope of existing research and presents valuable points. Secondly, it chooses a unique high-tech zone as the object of discussion, and empirically tests the relationship between population and public services system in the high-tech zone of Chengdu under the framework of sustainable development, in order to grasp the diversity of regional systems, to provide empirical evidence for promoting regional sustainable development as a whole.

## 2. Method

### 2.1. Data

The data in this paper mainly come from the relevant statistical data provided by the Public Security Bureau in the high-tech zone and the statistical yearbook in the high-tech zone of Chengdu from 2013 to 2022.

### 2.2. Method

2.2.1. Entropy Method

Step 1, normalize the initial data.
Treatment of positive indicators:

$$X_{ij} = \frac{X_{ij} - minX_j}{maxX_j - minX_j}$$

Treatment of negative indicators:

$$X_{ij} = \frac{maxX_j - X_{ij}}{maxX_j - minX_j}$$

Step 2, calculate the weight of item j of Variable i.

$$Y_{ij} = \frac{X_{ij}}{\sum X_{ij}}$$

Step 3, calculate the index information entropy.

$$E_j = -k\sum_{i=1}^{\alpha} Y_{ij} ln Y_{ij}$$

$$k = \frac{1}{ln\alpha}$$

Step 4, calculate the information entropy redundancy.

$$D_j = 1 - E_j$$

Step 5, calculate the indicator weights.

$$W_j = \frac{D_j}{\sum D_j}$$

Step 6, the comprehensive score of single indicator.

$$S_{ij} = W_j * X_{ij}$$

The entropy method is chosen to construct the evaluation indicator system of population and public services. Considering that the strategic core of the development in the high-tech zone of Chengdu is to respond to the uncertainty of the regional evolution with the strategy of flexible development, this paper uses the entropy method based on the difference-driven principle to obtain the weights of each indicator, to measure the development level of the population system and public services system [40]. Its applicability lies in the processing method of weighting by using the original data is more objective, and the importance of the entropy method to the discrete degree of the data elements is corresponding to the importance of the changing degree of the population conditions in the high-tech zone of Chengdu; the results presented by the use of the method are highly explanatory. It should be further explained that the use of the entropy method requires the standardization of data, that is, the conversion of indicators from absolute to relative values, which is intended to solve the problem of homogenization among heterogeneous indicators.

### 2.2.2. Coupling Degree Model

According to the concept and physical principle of the coupling degree model, the following equation can be established.

$$C_n = n \left[ \frac{U_1 U_2 \ldots \ldots U_n}{(U_1 + U_2 + \ldots \ldots + U_n)^n} \right]^{\frac{1}{n}}$$

Establish the coupling degree model of population system and public services system.

$$C = 2\sqrt{\frac{U_1 \times U_2}{(U_1 + U_2)^2}}$$

Among them, $U_i$ represents the evaluation value of system, C represents the coupling degree, $0 \leq C \leq 1$. When any evaluation value is 0, the coupling degree is 0; when all evaluation values are 0, the coupling degree is meaningless.

### 2.2.3. Coupling Coordination Degree Model

The degree of system coupling and coordination is reflected by the value of D, and the value of T is the comprehensive harmony index of the population system and public services system ($0 \leq T \leq 1$). $\alpha$ and $\beta$ are the weights of the population system and public services system, respectively.

$$D = \sqrt{CT}, \ 0 \leq D \leq 1$$

$$T = \alpha U_1 + \beta U_2, \ \alpha + \beta = 1$$

The synergy between two systems is very important for the change in order parameters from disorder to order, but focusing only on the coupling state may neglect the high-quality operation of the system. In order to explore the coordination between the population system and public services system, this paper introduces the criterion of coupling coordination degree [41]; the coupling degree and the coupling coordination degree in the high-tech zone of Chengdu are calculated based on the positive properties of the above-mentioned indexes [42], in order to further explore its element penetration, connectivity level and co-evolution characteristics.

### 2.3. Procedure

This paper intends to obey the following process to promote research (Figure 1).

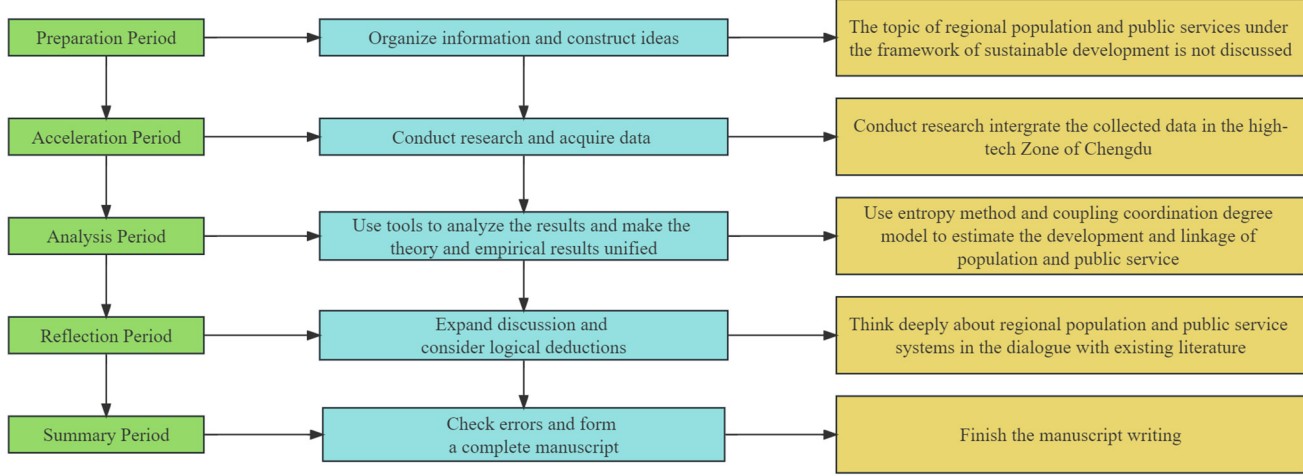

**Figure 1.** Research flow.

## 3. Results

### 3.1. Population System

The population system is a general term to analyze the object of population from the perspective of systems theory. The population system can be divided into the internal population system and the external population system. The internal population system is concerned with the birth, death, migration, mobility and other development characteristics of the population, while the external population system is concerned with the connection between the population and the surrounding environment, such as the formation of economic, cultural and other elements of the network. According to the reality in the high-tech zone of Chengdu, the birth effect and death effect of the internal population system are counteracted at a macro level; the population distribution and population quality have an obvious influence on regional sustainable development. Using population density to measure population distribution can reflect the support effect of population spatial distribution on regional sustainable development. Using the per capita education years aged 15 and above to measure population quality can reflect the supporting effect of population quality on regional sustainable development to the great extent. These two indicators are helpful to present the evolution and present situation of the internal population system in the high-tech zone of Chengdu, and also provide a valuable reference for monitoring the internal population system and adjusting the development policy in time. In the external population system, the economic system is the most obvious one which is related to population. Using per capita disposable income of urban residents to measure the state of economic development can best reflect the success and vitality of the external population system in regional sustainable development, and it can also provide evidential support for monitoring the external population system and changing the strategic focus in time, which is beneficial for presenting the evolution and present situation of the external population system in the high-tech zone of Chengdu. The specific indicator systems are shown in Table 1.

**Table 1.** Population indicator system in the high-tech zone of Chengdu.

| Indicator | Second Indicator | Third Level Indicator | Indicator Type | Indicator Weight |
|-----------|------------------|----------------------|----------------|------------------|
| | Population size | Population density | Intensity indicator | 0.310 |
| Population | Population quality | Per capita education years aged 15 and above | Intensity indicator | 0.379 |
| | Population welfare | Per capita disposable income of urban residents | Intensity indicator | 0.311 |

The overall development level of the population system in the high-tech zone of Chengdu shows a positive evolutionary trend (Figure 2). From the perspective of time trend, using the average development level of population system evaluation results as the standard, the population system development in the high-tech zone of Chengdu can be divided into two stages.

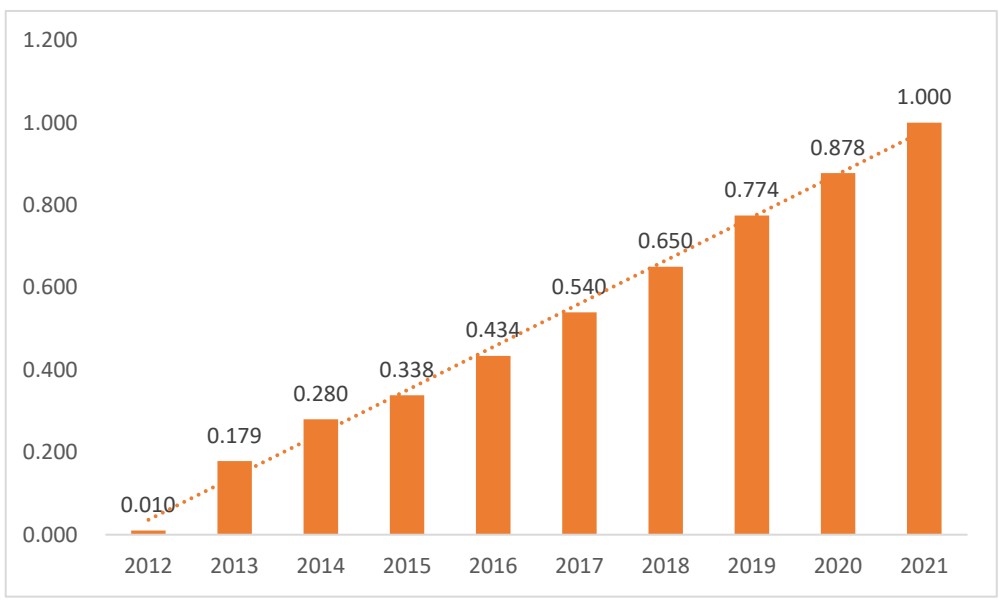

**Figure 2.** Population system evaluation results in the high-tech zone of Chengdu.

The first stage is the advantages accumulation stage of the population system (2012–2016). Although the evaluation results of the population system in this stage are lower than the average at the whole data level, the development foundation of the population system in this stage has been greatly consolidated. Upon examining the development context of this stage, both the internal and external population systems in the high-tech zone of Chengdu are orderly accumulating value while fully respecting the characteristics and regulations. On the one hand, the high-tech zone of Chengdu has continuously attracted a floating or migrating population and even talents gathered through reasonable policies such as the "Implementation Measures for Encouraging High level Talents to Enter the High-Tech Zone of Chengdu for Innovation and Entrepreneurship" and the "Implementation Rules for Introducing High; Level Talents into the High-Tech Zone of Chengdu for Entrepreneurship", thereby increasing population density and consolidating the support of the internal population system for regional sustainable development. At the same time, the high-tech zone of Chengdu has focused on increasing the integration of educational resources, improving the allocation effect of educational resources, and continuously improving the inclusiveness of educational development. On the other hand, the high-tech zone of Chengdu actively explores emerging economic growth points and strives to find development paths that are suitable for local conditions to drive better and faster employment of the regional population, thereby increasing the per capita disposable income of residents and promoting economic growth of the external population system.

The second stage is the potential release stage of the population system (2017–2021), in which the population system evaluation results are higher than the average at the whole data level, and the previous accumulation of the population system is effectively released during this stage. Upon examining the development context of this stage, both the internal and external population systems in the high-tech zone of Chengdu are continuously promoting the realization of a new pattern of regional population development while fully respecting the characteristics and regulations. On the one hand, the high-tech zone of Chengdu has implemented a policy of a floating or migrating population, continuously improving the construction of supporting facilities for the population's talents, and gradually

realizing the transformation from population introduction to population retention. This transformation has prompted the population agglomeration effect in the high-tech zone of Chengdu to shift towards a population demonstration effect. The increase in population density in the high-tech zone of Chengdu under the population demonstration effect is no longer only manifested as the deepening of image colors at the geographical spatial level, but also as the deepening of the impact of the internal population system on the comprehensive efficiency of regional sustainable development and the realization of the value diffusion of the internal population system. On the other hand, the high-tech zone of Chengdu fully grasps the essence of high-quality economic development, guided by the stable employment needs, and takes multiple measures to solve the problem of an non-working local eligible labor force, thereby increasing the per capita disposable income of residents. At the same time, the high-tech zone of Chengdu fully grasps the lag effect of talent dividends, ensures the steady increase in the share of talent contributions in economic growth and continuously promotes the stabilization and optimization of the economic growth momentum of the external population system.

Combined with the indicator weights, the population density, per capita education years aged 15 and above and per capita disposable income of urban residents within the population indicator system in the high-tech zone of Chengdu are 0.310, 0.379 and 0.311, respectively, with little difference among the three indicators. This indicates that, firstly, both the internal population system in the high-tech zone of Chengdu, characterized by population density and per capita education years of 15 years and above, and the external population system in the high-tech zone of Chengdu, characterized by per capita disposable income of urban residents, have achieved optimization to varying degrees. Secondly, the internal population system characterized by population density and per capita education years of 15 years and above, as well as the external population system characterized by the per capita disposable income of urban residents, have made positive contributions to the positive evolution of the population system development in the high-tech zone of Chengdu.

### 3.2. Public Services System

The public services system is a general term for analyzing the value of public goods and services provided by public organizations with a common consumption from the perspective of systems theory. The public services system can be divided into different subsystems according to different standards, such as education, healthcare, social security, etc. Based on the status in the high-tech zone of Chengdu, its public services system is relatively complete, and the attention of the government and the public to the overall public services is also continuously increasing. Social security, education, healthcare, culture, transportation, innovation, environment and digitalization are eight key public services subsystems in the high-tech zone of Chengdu with high representativeness, which can be beneficial from the viewpoint of regional sustainable development. Therefore, this paper observes the development of public services from eight subsystems. At the same time, this paper comprehensively uses scale indicators, structural indicators and intensity indicators to measure the development level of the public services system. Its rationality lies in the fact that the scale indicators represented by the number of urban residents participating in unemployment insurance can reflect the specific situation at the macro level, thereby demonstrating the effectiveness of policies and investments; the structural indicators represented by the proportion of education expenditure can reflect the intensity of attention and investment design, thus indicating the bias of policies and investments; the intensity index represented by effective invention patents per 10,000 permanent residents can reflect the sense of development acquisition, thereby demonstrating the conversion effect of policies and investments. The above parts are selected using standards which are positive for regional sustainable development. The specific indicator systems are shown in Table 2.

**Table 2.** Public services indicator system in the high-tech zone of Chengdu.

| Indicator | Second Indicator | Third Level Indicator | Indicator Type | Indicator Weight |
|---|---|---|---|---|
| Public services | Social security services | Number of urban residents participating in unemployment insurance | Scale indicator | 0.195 |
| | Education services | Education expenditure/Total fiscal expenditure | Structural indicator | 0.077 |
| | Healthcare services | Number of beds in medical institutions per 10,000 permanent residents | Intensity indicator | 0.143 |
| | Culture services | Paper book ownership per 10,000 permanent residents | Intensity indicator | 0.155 |
| | Transportation services | Transportation expenditure/Total fiscal expenditure | Structural indicator | 0.103 |
| | Innovation services | Effective invention patents per 10,000 permanent residents | Intensity indicator | 0.169 |
| | Environment services | Green coverage rate in built-up areas | Structural indicator | 0.065 |
| | Digitalization services | Number of registered digital TV users | Scale indicator | 0.093 |

The overall development level of the public services system in the high-tech zone of Chengdu shows a fluctuating evolutionary trend (Figure 3). From the perspective of the time trend, the development of the public services system in the high-tech zone of Chengdu can be roughly divided into two stages according to the permanent residents (Figure 4). In the old norm stage of public services system, the number of permanent residents in the high-tech zone of Chengdu remained at 0.5 million, while in the new norm stage of the public services system, the number of permanent residents in the high-tech zone of Chengdu reached or even exceeded 1 million. The increase in population is enough to shape the state and pattern of public services system.

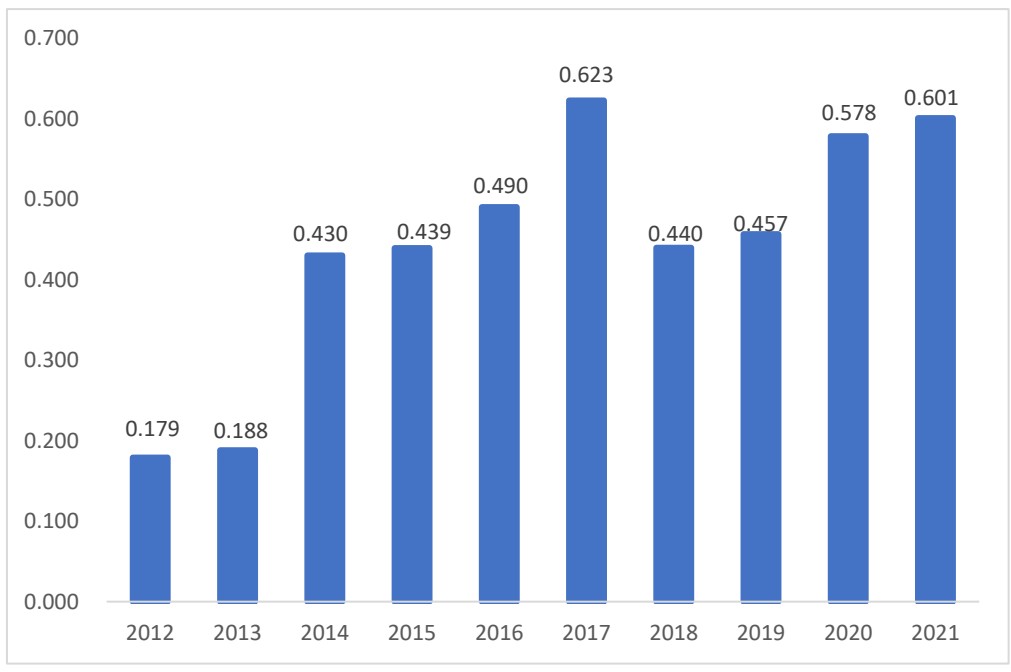

**Figure 3.** Public services system evaluation results in the high-tech zone of Chengdu.

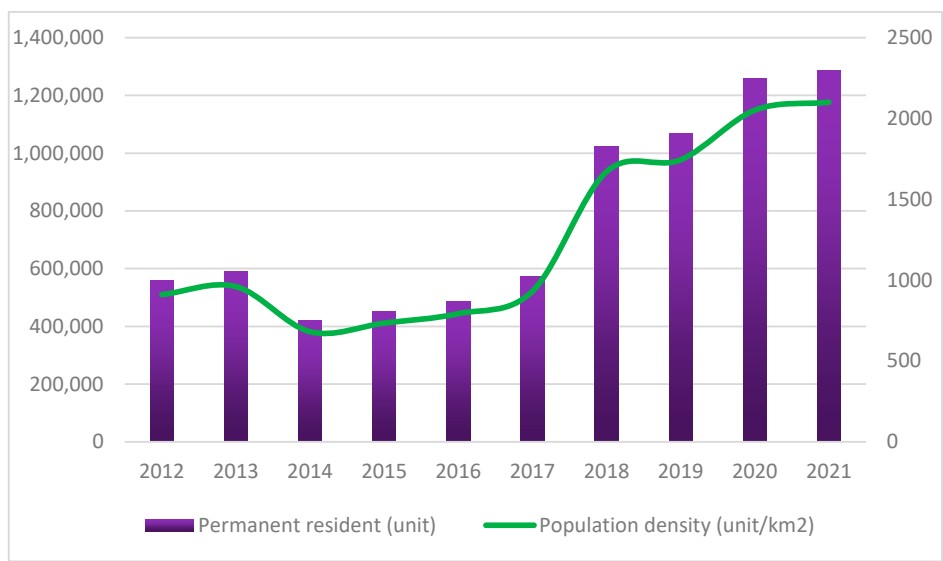

**Figure 4.** Permanent resident and population density in the high-tech zone of Chengdu.

The first stage is the old norm stage of the public services system (2012–2017). During the process from the evaluation results of the public services system in 2012 (0.179) to the evaluation results of the public services system in 2017 (0.623), the improvement in the level of the public services system in the high-tech zone of Chengdu is the result of the complex interweaving of development and change. The development is mainly reflected in the improvement in social security services level, innovation services level, environmental services level and digital services level. The changes are mainly manifested in the phased stagnation in the improvement in education services level, transportation services level, healthcare services level and culture services level. During this period, the population in the high-tech zone of Chengdu remained at around 0.5 million. The changing characteristics of public services in the high-tech zone of Chengdu are closely related to this background, that is, the steady improvement in public services level is to some extent attributed to the stability of population.

In the year from the evaluation results of the public services system in 2017 (0.623) to the evaluation results of the public services system in 2018 (0.440), the rapid decline in the level of public services in the high-tech zone of Chengdu is mainly due to the surge in the number of permanent residents, reflected in the sudden decline in the level of innovation services, healthcare services and culture services in the public services system. This indicates that the rapid increase in population has a significant inhibitory effect on the improvement of public services level in the high-tech zone of Chengdu in the short term.

The second stage is the new norm stage of the public services system (2018–2021). In the process from the evaluation results of the public services system in 2018 (0.440) to the evaluation results of the public services system in 2021 (0.601), the improvement in the level of the public services system in the high-tech zone of Chengdu is the result of actively adapting to the new characteristics of population development. The inertia of actively adapting to the new population development is mainly reflected in the continuous improvement in social security services level manifested in the improvement in the ability to provide basic support, the innovation services level manifested in a full scale improvement in innovation input, innovation capability, and innovation output, environment services level manifested in the improvement of environmental inclusion and digital services level manifested in the improvement in people's livelihood embedding level. The changes are mainly manifested in the phased increase in the level of education services manifested in the enhancement of attention to education services and the effective transformation of investment in education development reform, and transportation services level manifested in the slight fluctuations in the proportion of transportation expenditure. The dilemma

is mainly reflected in the phased problem of improving the level of healthcare services manifested in the increase in the probability of periodic medical runs on the services occurring, and culture services manifested in the weakening of the support effect of individual cultural resource management during this period. During this period, the permanent population in the high-tech zone of Chengdu exceeded the one million level, becoming a population residential area with new characteristics. The changing characteristics of public services composition indicators in the high-tech zone of Chengdu are closely related to the background of the population. The steady improvement in public services level requires systematic consideration of the development impact brought by population changes, active adaptation to population changes and active resolution of development pain points. This is currently a good strategy to improve the public services level in the high-tech zone of Chengdu.

Based on the weight of the indicators, the highest weight in the public services system in the high-tech zone of Chengdu is the number of urban residents participating in unemployment insurance (0.195), while the lowest weight is the green coverage rate of the built-up area (0.065), with little difference in weight between indicators. This indicates that, firstly, the four dimensions of social security services, innovation services, environment services, and digital services have made sustained and positive contributions to the improvement in the public services system level in the high-tech zone of Chengdu. Secondly, healthcare and culture services have to some extent hindered the positive evolution of the public services system in the high-tech zone of Chengdu.

### 3.3. Coupling and Coordination Effect of Population and Public Services System

As is shown in Figure 5 and Table 3, firstly, from 2012 to 2021, the coupling degree between the population system and public services system in the high-tech zone of Chengdu has always maintained a high level, indicating that there is a high degree of connection between the two systems in the high-tech zone of Chengdu, and there is a situation of factor infiltration and factor network connection between two systems. Secondly, there are strong differences in the coordination degree and coupling coordination degree between the population system and public services system in in the high-tech zone of Chengdu from 2012 to 2021. From the perspective of time, the coordination degree and coupling coordination degree between the population system and public services system in the high-tech zone of Chengdu generally show an increasing trend over time during the sample period. The coupling coordination degree increased from 0.100 (2012) to 0.983 (2021), with an overall increase of up to 883%. Observing the coupling coordination effect, the coordination ratio is greater than the imbalance ratio, and the types of coordination and imbalance are relatively stable. The coupling coordination effect has undergone a gradual transition from the status of severe imbalance to the status of high-quality coordination. These results fully indicate that the coupling and coordination effect between the population and public services system in the high-tech zone of Chengdu has a gradually optimized evolutionary characteristic. Thirdly, the development pace of the population system is slightly faster than that of the public services system. In the early stage of the research sample (2012–2016), the effectiveness of the population system is relatively high, while the effectiveness of the public services is relatively low. The population system have a traction effect on the public services system. In the later stage of the research sample (2017–2021), the development of public services has a certain sense of catching up, and various services subsystems actively adapt to changes in regional sustainable development patterns and conditions. Without compromising structural order, various services subsystems focus on optimizing services strategies and enriching services content, thereby promoting the optimization of public services system; narrowing the differences between the population system and public services system; realizing positive feedback between the population and public services system. Fourthly, the change in the coupling and coordination effect between the population system and public services system reflects certain characteristics of the development in the high-tech zone of Chengdu. The first characteristic is that the

amount of population in the high-tech zone of Chengdu plays a very important role in the public services system, and has a significant impact on relevant factors. The second characteristic is that the development of the public services system belongs to the weakness under the framework of regional sustainable development. Furthermore, not only has the improvement in the level of the public services system intrinsic value, but also the improvement has external diffusion effects.

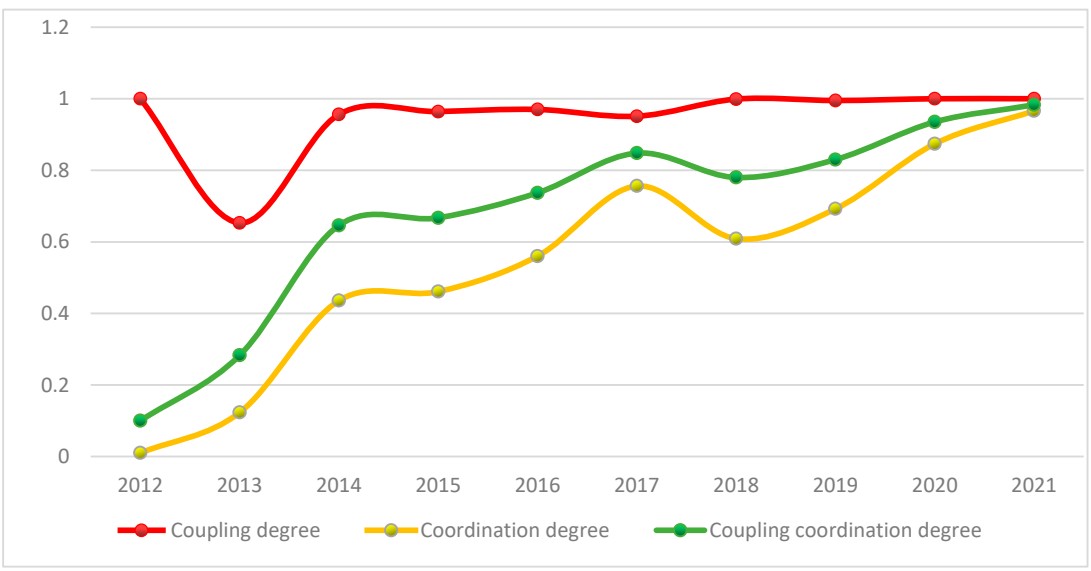

**Figure 5.** Coupling and coordination effect of two systems in the high-tech zone of Chengdu.

**Table 3.** Coupling and coordination effect of two systems in the high-tech zone of Chengdu.

| Time | Coupling Degree | Coordination Degree | Coupling Coordination Degree | Effect |
|------|------|------|------|------|
| 2012 | 1.000 | 0.010 | 0.100 | Severe imbalance |
| 2013 | 0.653 | 0.123 | 0.283 | Moderate imbalance |
| 2014 | 0.956 | 0.436 | 0.646 | Primary coordination |
| 2015 | 0.964 | 0.461 | 0.667 | Primary coordination |
| 2016 | 0.970 | 0.560 | 0.737 | Intermediate coordination |
| 2017 | 0.951 | 0.756 | 0.848 | Good coordination |
| 2018 | 0.999 | 0.609 | 0.780 | Intermediate coordination |
| 2019 | 0.995 | 0.692 | 0.830 | Good coordination |
| 2020 | 1.000 | 0.874 | 0.935 | High-quality coordination |
| 2021 | 1.000 | 0.966 | 0.983 | High-quality coordination |

## 4. Discussion

From the perspective of the research findings, firstly, the development level of the population system in the high-tech zone of Chengdu shows a positive evolutionary trend in the sample period, and the population system development has a stable contribution to the sustainable development in the high-tech zone of Chengdu. Secondly, the development level of the public services system in the high-tech zone of Chengdu shows a dynamic evolutionary trend in the sample period. The contribution of the public services system to sustainable development in the high-tech zone of Chengdu is affected by the population system. Thirdly, the correlation degree between the population and public services system in the high-tech zone of Chengdu is kept at a relatively high level during the sample period, and the two subsystems have a relatively close element fusion. At the same time, the effect between population and public services system is optimized in the sample period. There are positive information exchanges, value feedback and dynamic adjustments between the two subsystems.

From the perspective of value extension of the research findings, firstly, based on the conclusion of the population system, we can find that there is still a lack of attention to the structural changes of population development. The concrete manifestations are as follows:

(1) The strength of the disclosure of the population age structure data is insufficient, which is similar to the findings of the previous research [43].

(2) The employment structure of the population still has a large adjustment space, and the contribution of the working population in the secondary and tertiary industries is not significant in the economic growth system, which shows some deviation from the conclusion of the previous research [44].

(3) The structural effect of talent allocation is still lacking. After the introduction of high-end talents, there is still a certain decoupling between the allocation effect and the high-quality development of regional economy, and there is a space for improvement in the transformation mechanism from human capital of basic talents to talents dividend. This greatly expanded the previous research on the demographic dividend [45], so that the population dividend in a similar high-tech zone ought to concentrate on the performance of talent dividend.

Secondly, based on the discussion of the population system in the high-tech zone of Chengdu, we can find that there are obvious shortcomings in the public services system in the high-tech zone of Chengdu. The concrete manifestations are as follows:

(1) The guaranteed abilities of medical resources have certain deficiencies. Specifically, compared with other high-tech zones in the eastern and central regions, the difference in the total number of beds per 10,000 permanent residents in medical and health institutions is not obvious, but there is a huge gap between the rationality of distribution and the sense of access to medical resources. Compared with the existing research results [46], there is still a long way to go for the development of public services in the high-tech zone of Chengdu.

(2) The supply effect of cultural resources can be improved greatly. In particular, compared with other high-tech zones in the eastern and central regions, there is no great difference in the reserve of cultural resources, but the view of the allocation of cultural resources is still in a relatively passive supply stage, and the supply is not systematic, so that causing the reverse development of supply from adequate reserves to poor evaluation [47]. The problem of supply effectiveness needs to be solved urgently [48].

Thirdly, based on the conclusion of the connection between the population system and public services system in the high-tech zone of Chengdu, what can be found is that there are some communication barriers and transfer shackles between the population system and public services system in the high-tech zone of Chengdu. The concrete manifestations are as follows:

(1) There is a decoupling between the population system and the coping capacity of the public services system, which occurs occasionally in China [49].

(2) There is a difference between the population system quality improvement and the satisfaction ability of public services system and the development of public services system lags behind the development of population system. The above cognition together with the existing research results, proves the non-synchronization between systems [50].

From the perspective of research process, this paper has certain shortcomings. First of all, the measurement of the population system is not complete, and there is a further rich space for the external population system of consideration. Secondly, the sample size of this study on the high-tech zone in western China is slightly insufficient, and the reference value of the conclusions drawn from this study needs further observation. Finally, the contribution coefficients of population and public services to the sustainable development in the high-tech zone are not estimated in the article, which merit further discussion in future research.

From the perspective of future research, this paper has the following expectations. Firstly, referring to the existing research [51], we can bring the element of the external population system, such as resource and environmental factors, into the population de-

velopment evaluation system under the condition of increasing the availability of data; therefore, the results of the population system development measurement are closer to reality. Additionally, broaden the range of sample selection. This research provided us with the value reference [52]. If the conditions permit, we can try to bring more high-tech areas into the research object, so as to obtain more universal value enlightenment. Finally, an empirical study on the topic of the contribution of population and public services to regional sustainable development can be conducted. For example, based on the relevant research [53], future research can discuss the causality among the objects of study using the VAR model. Based on the relevant research [54], the panel model can be used to discuss the effect of population and public services on the level of regional sustainable development. Based on the relevant research [55], future research can discuss the population- and public services-related effects on the level of regional sustainable development with the help of a spatial model.

## 5. Conclusions and Implication

### 5.1. Conclusions

From the perspective of research findings, firstly, the development level of the population system in the high-tech zone of Chengdu shows a positive evolutionary trend in the sample period, and the population system development makes a stable contribution to the sustainable development in the high-tech zone of Chengdu. Secondly, the development level of public services system in the high-tech zone of Chengdu shows a dynamic evolutionary trend in the sample period. The contribution of public services system to the sustainable development in the high-tech zone of Chengdu is affected by the population system. Thirdly, the correlation degree between the population and public services system in the high-tech zone of Chengdu is kept at a relatively high level during the sample period, and the two subsystems have a relatively close element fusion. At the same time, the effect between the population and public services system is optimized in the sample period. There are positive information exchanges, value feedback and dynamic adjustments between the two subsystems.

### 5.2. Implications

The high-tech zone of Chengdu needs to attach adequate importance to the following aspects. Firstly, optimize the distribution of medical resources and improve the quality of medical services to meet the growing population demand. Government departments should increase investment in medical resources, increase the number of beds in medical and health institutions per 10,000 permanent residents and pay attention to the rationality of resource distribution to improve residents' satisfaction with medical treatment. Secondly, improve the supply level of cultural resource management, pay attention to the effective and systematic supply of cultural resource management, and enhance residents' sense of access to cultural resource management. Government departments should obey the orientation of population demand, optimize the allocation of cultural resource management, improve the systematic and effective supply of cultural resource management, and meet the diverse cultural needs of residents. Thirdly, establish effective channels for information communication and coordinate the relationship between the population system and public services system. By utilizing information technology, information sharing and connection between the population system and public services system can be achieved, addressing the decoupling phenomenon between the surge in population systems and the response capabilities of public services systems. Fourthly, pay attention to changes in population structure and demand, and formulate corresponding policies to narrow the gap between the improvement of population system quality and the satisfaction ability of public services system. Government departments should strengthen the coordination between structural changes in the population system and structural adjustments in the public services system, in order to achieve synchronization and coordination between speed and effectiveness. Fifthly, increase investment, improve the development level of the public services sys-

tem and narrow the gap between the development of the population system and public services system.

**Author Contributions:** X.W.: conceptualization, Methodology. T.H.: data curation, writing—review and editing. K.L.: data Curation, writing—review and editing. All authors have read and agreed to the published version of the manuscript.

**Funding:** This work was supported by the Key Project of the National Social Science Fund, China (21ARK001); the Fundamental Research Funds for the Central Universities, China, (JBK2304087); the Fundamental Research Funds for the Central Universities, China, (JBK230110).

**Institutional Review Board Statement:** Not applicable.

**Informed Consent Statement:** Not applicable.

**Data Availability Statement:** The data used to support the results of this study are available from the corresponding author upon request.

**Conflicts of Interest:** The authors declare no conflict of interest.

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
