# Peer review of "Regional Population and Public Services under the Framework of Sustainable Development: Evidence from a Typical High-Tech Zone in China"

_sustainability, doi:10.3390/su151713259_

Round 1

Reviewer 1 Report

From the perspective of sustainable development, based on entropy model and coupling coordination model, this manuscript analyzes the synergistic relationship between population system and public service system in China Western High-tech Zone and its contribution to sustainable development. The overall idea of the article is clear, but the conclusion lacks sufficient experimental support, and there are few overall charts, lengthy analysis and lack of geospatial information expression. It is suggested to add experimental contents and charts to simplify the analysis.

Other minor issues are as follows:

1) The text analysis of Section 3 "Public Service System" of the manuscript is too lengthy, so it is suggested to simplify and split this part and add supporting contents such as charts.

2) There are some problems with the indentation format of some paragraphs in the manuscript, such as Section 1, Paragraph 2 and Section 4, Paragraph 5.

3) Section 4 of the manuscript suggests subsection to explain the general situation of Chengdu and the shortcomings of the experiment respectively.

None

Author Response

Response to the Reviewers’ Comments on the PaperSustainability-2576328Entitled“Regional Population and Public Services under the Framework of Sustainable Development: Evidence from Typical High-Tech Zone in China”

August 23, 2023

We would like to thank you for your efforts in reviewing our manuscript and providing many helpful comments and suggestions, which will all prove invaluable in the revision and improvement of our paper, as well as in guiding our research in the future .To facilitate this discussion, we first retype your comments in italic font and then present our responses to the comments.

Response to reviewer #1: 

1.The overall idea of the article is clear, but the conclusion lacks sufficient experimental support, and there are few overall charts, lengthy analysis and lack of geospatial information expression. It is suggested to add experimental contents and charts to simplify the analysis.

  • We feel great thanks for your professional review work on our article. As you are concerned, there are several problems setting to be addressed. According to your nice suggestions, we have made extensive adjustment to our previous draft. First, remove the lengthy analysis of the public service system in the Section 3. Second, optimize  the public service system analysis ideas and statements in the Section 3. Third, the time trend chart of the standard of dividing old norm and new norm is added, which is helpful to grasp the change of resident population in the high-tech zone of Chengdu City.

2.The text analysis of Section 3 "Public Service System" of the manuscript is too lengthy, so it is suggested to simplify and split this part and add supporting contents such as charts.

  • Thanks for your precious suggestion. We have tried our best to simplify the content of relevant section.The changes to the manuscript are giving in the blue text, and we add a new chart.  

The first stage is the old norm stage of the public services system (2012-2017). During the process from the evaluation result of the public services system in 2012 (0.179) to the evaluation result of the public services system in 2017 (0.623), the improvement of the level of the public services system in the high-tech zone of Chengdu City is the result of the complex interweaving of development and change. The development is mainly reflected in the improvement of social security services level, innovation services level, environment services level, and digital services level. The changes are mainly manifested in the phased stagnation in the improvement of education services level, transportation services level, healthcare services level and culture services level. During this period, the population of the high-tech zone of Chengdu City remained at around 0.5 million. The changing characteristics of public services in the high-tech zone of Chengdu City are closely related to this background, that is, the steady improvement of public services level is to some extent attributed to the stability of population.

The second stage is the new norm stage of the public services system (2018-2021). In the process from the evaluation result of the public services system in 2018 (0.440) to the evaluation result of the public services system in 2021 (0.601), the improvement of the level of the public services system in the high-tech zone of Chengdu City is the result of actively adapting to the new characteristics of population development. The inertia of actively adapting to the new population development is mainly reflected in the continuous improvement of social security services level manifested in the improvement of the ability to provide basic support, innovation services level manifested in a full chain improvement of innovation input, innovation capability, and innovation output, environment services level manifested in the improvement of environmental inclusion, and digital services level manifested in the improvement of people's livelihood embedding level. The changes are mainly manifested in the phased increase in the level of education services manifested in the enhancement of attention to education services and the effective transformation of investment in education development reform, and transportation services level manifested in the slight fluctuations in the proportion of transportation expenditure. The dilemma is mainly reflected in the phased problem of improving the level of healthcare services manifested in the increase in the probability of periodic medical runs occurring, and culture services manifested in the weakening of the support effect of individual cultural resource management during this period. During this period, the permanent population of the high-tech zone of Chengdu City exceeded the 1 million level, becoming a population residential area with new characteristics. The changing characteristics of public services composition indicators in the high-tech zone of Chengdu City are closely related to the background of population. The steady improvement of public services level requires systematic consideration of the development impact brought by population changes, active adaptation to population changes, and active resolution of development pain points. This is currently a good strategy to improve the public services level of the high-tech zone of Chengdu City.           

From the perspective of time trend, the development of the public services system of the high-tech zone of Chengdu City can be roughly divided into two stages according to the permanent residents (Figure 4).

Figure 4. Permanent resident and population density of the high-tech zone of Chengdu City.

3.There are some problems with the indentation format of some paragraphs in the manuscript, such as Section 1, Paragraph 2 and Section 4, Paragraph 5.

  • We sincerely thank the reviewer for careful reading. As suggested by the reviewer, we have corrected the indentation

4.Section 4 of the manuscript suggests subsection to explain the general situation of Chengdu and the shortcomings of the experiment respectively.

  • Thanks for your precious suggestion. We haveexplained the general situation of the high-tech zone of Chengdu City and rewritten the shortcomings of the experiment.The changes to the manuscript are giving in the blue text, and we add a new chart.  

First of all, the high-tech zone of Chengdu City has the main characteristics of other high-tech zones, such as large population, large floating population, high degree of industrial agglomeration, relatively large scientific and technological talents, high population quality, relatively young population structure, strong support of digital innovation, and relatively clear policy bias (Xie et al., 2018; Gong et al., 2023; Liu et al., 2023; Li et al., 2023; Coccia, 2023; Hua, & Hu, 2023). Secondly, the high-tech zone of Chengdu City is the driving force of the development of science and technology in the Western Region of China, and plays an exemplary role in the process of building the regional economic growth pole and realizing the radiation effect on the external economy (Du et al., 2022). For example, reducing the degree of resource mismatch and improving total factor production (Jin et al., 2022; Zeng et al., 2022). Compared with other high-tech zones, the irreplaceable feature of the high-tech zone of Chengdu City is more significant.

From the perspective of research process, this paper has certain shortcomings. First of all, the measurement of population system is not complete, and there is a further rich space for the population of the external system of consideration. Secondly, the sample size of this study on the western high-tech zone is slightly insufficient, and the reference value of the conclusions drawn from this study needs further observation. Finally, the contribution coefficients of population and public services to the sustainable development of high-tech zones are not estimated in the article, which worth further discussion in future research.

Reviewer 2 Report

The manuscript develops a significant argument and contribution which is worth publishing. However, some revisions are still required to make it publishable based on the following comments.

1.       Please highlight the novelty of your research. What makes it distinguishable compared to previously conducted studies in this field?

2.       Provide a concise research implication in the last part of your abstract.

3.       The authors need to enrich the reviewed literature in the field strikingly and systematically.

4.       The authors must formulate research questions in the form of interrogative sentences.

5.       The methods section is the weakest part of this paper by far. It requires further nuanced details and descriptions regarding the procedure of implementing the research. It needs to be organized systematically. Different parts need to be sub-categorized in the methods section.

6.       The research flow should be organized and illustrated in the form of a diagram.

7.       Paragraphing needs to be reviewed and restructured.

8.       What are the limitations associated with this study?

9.       What are your suggestions for the future research agenda in this field?

Author Response

Response to the Reviewers’ Comments on the PaperSustainability-2576328Entitled“Regional Population and Public Services under the Framework of Sustainable Development: Evidence from Typical High-Tech Zone in China”

August 22, 2023

We would like to thank you for your efforts in reviewing our manuscript and providing many helpful comments and suggestions, which will all prove invaluable in the revision and improvement of our paper, as well as in guiding our research in the future .To facilitate this discussion, we first retype your comments in italic font and then present our responses to the comments.

Response to reviewer #2

  1. Please highlight the novelty of your research. What makes it distinguishable compared to previously conducted studies in this field?
  • Thanks for your precious suggestion. According to your nice suggestion, we have added the concise research implication in the last part of abstract. The changes to the manuscript are giving in the blue text.

The novelty of this paper lies in: firstly, the system theory is introduced as the cognitive perspective of sustainable development, which broadens the scope of existing research and presents valuable points. Secondly, it chooses the unique high-tech zone as the object of discussion, and empirically tests the relationship between population and public services system in the high-tech zone of Chengdu City under the framework of sustainable development, in order to grasp the diversity of regional system, to provide empirical evidence for promoting regional sustainable development as a whole.

  1. 2.Provide a concise research implication in the last part of your abstract.
  • Thanks for your precious suggestion. According to your nice suggestion, we have added the concise research implication in the last part of abstract.The changes to the manuscript are giving in the blue text.

The research implication is dialectically grasp the development and linkage between regional population and public services system under the framework of sustainable development.   

  1. 3.The authors need to enrich the reviewed literature in the field strikingly and systematically.
  • Thanks for your precious suggestion. Owing thatthe literature review and research issues are closely related, so we do not put the literature review in a single chapter, so please forgive the inconvenience caused to your reading. In light of your comments, we have added a number of transitional cues to the literature review section and reviewed the relationship between population and public services on an ongoing basis.

Population system is the general term of value to analyze the object of population, which can be divided into internal system and external system (Barbara, 2021). The sustainability of population systems is usually manifested in the optimization of the spatial distribution of the population, the improvement of the educational level of the population, and the enhancement of the well-being of the population (Abramovich & Vasiliu, 2022; Nishita et al., 2022; Shi et al., 2023; Song & Yuan, 2022). At the same time, the public services system is the general term of value to analyze the public goods and the services which the public organization provides has the common consumption nature the value general name, and it covers goods, services, organizations etc (Witesman et al., 2023; Wang et al., 2023). The sustainability of public services is reflected in the increase of educational resources (Gray et al., 2023), the optimization of traffic conditions (Tiglao et al., 2023; RambaldiniGooding et al., 2022), the health care system (Qiu et al., 2023), the cultural level of the leap (Zhang et al., 2023), the consolidation of social security functions (Duggan et al., 2023), and the improvement of digital services capacity (Sabrina et al., 2023). The value contribution of population and public services system in regional system shows that the former provides dynamic support for sustainable development of regional system, while the latter provides process guarantee for sustainable development of regional system (Nakray, 2022).

The value of population and public services systems lies not only in their independence but also in their interaction (Zhang et al., 2023). In China, for example, the early public services in China was linked to the household registration system. With the expansion and acceleration of population migration, the supply of public services are facing the disconnection between scale and quality, rational allocation of regional resources and sustainable development are facing great challenges (Pan et al., 2022). Therefore, the effect and intensity of population and public services are of great value to the achievement of sustainable development.

  1. The authors must formulate research questions in the form of interrogative sentences.
  • Thanks for your precious suggestion. As you are concerned, theresearch questions ought to be proposed in the form of interrogative sentences. So we have adjusted the relevant expression. The changes to the manuscript are giving in the blue text.

Based on the literature review, the two problems deserve to be discussed in depth. Firstly, what is the level of population and public service development in the typical high-tech zone in western China? Secondly, what is the relationship between the population and public service development in the typical high-tech zone in western China? In order to reply to the mentioned problem, this paper aims to analyze the population and public services system of the typical high-tech zone in western China under the framework of sustainable development.

  1. The methods section is the weakest part of this paper by far. It requires further nuanced details and descriptions regarding the procedure of implementing the research. It needs to be organized systematically. Different parts need to be sub-categorized in the methods section.
  • Thanks for your precious suggestion. As you are concerned, the section of method ought to be adjusted. So we have rewritten the section of method. The changes to the manuscript are giving in the blue text.
  1. METHOD

2.1. Data

the data in this paper mainly come from the relevant statistical data provided by the Public Security Bureau of high-tech zone and the statistical yearbook of high-tech zone of Chengdu City from 2013 to 2022.

2.2 Method

2.2.1 Entropy method

Sep1, Normalize the initial data.

Treatment of positive indicators:

Treatment of negative indicators:

Step 2, Calculate the weight of item j of Variable i.

Step 3, Calculate the index information entropy.

   Step 4, Calculate the information entropy redundancy.

Step 5, Calculate the indicator weights.

Step 6, The comprehensive score of single indicator.

Choosing the entropy method to construct the evaluation indicator system of population system and public services system. Considering that the strategic core of the development of the high-tech zone of Chengdu city is to respond to the uncertainty of the regional evolution with the strategy of flexible development, this paper uses the entropy method based on the difference-driven principle to obtain the weights of each index, to measure the development level of population system and public services system (Zare, 2023). Its applicability lies in: the processing method of weighting by using the original data is more objective, and the importance of the entropy method to the discrete degree of the data elements is corresponding to the importance of the changing degree of the population condition in the high-tech zone of Chengdu City, the results presented by the use of method are highly explanatory. It should be further explained that the use of the entropy method requires the standardization of data, that is, the conversion of indicators from absolute to relative values, which is intended to solve the problem of homogenization among heterogeneous indicators.

2.2.2Coupling degree model

According to the concept and physical principle of coupling degree model, the following equation can be established.

Establish the coupling degree model of population system and public services system.

Among them, represents the evaluation value of system, C represents the coupling degree, 0≤C≤1. When any evaluation value is 0, the coupling degree is 0; when all evaluation values are 0, the coupling degree is meaningless.

2.2.3Coupling coordination degree model

                  , 0≤D≤1

                   ,

Degree of system coupling and coordination is reflected by the value of D, and the value of T is the comprehensive harmony index of the population system and public services system (0≤T≤1). α and β are the weights of the population system and public services system, respectively.

 The synergy between systems is very important for the change of order parameters from disorder to order, but focusing only on the coupling state may neglect the high-quality operation of the system. In order to explore the coordination between population system and public services system, this paper introduces the criterion of coupling coordination degree (Zhang et al., 2023), the coupling degree and the coupling coordination degree of the high-tech Zone of Chengdu City are calculated based on the positive property of the above-mentioned indexes (Cheng et al., 2023), in order to further explore its element penetration, connectivity level and co-evolution characteristics.

  1. 6.The research flow should be organized and illustrated in the form of a diagram.
  • Thanks for your precious suggestion. As you are stated, We use the form of diagrams to show the research flow, which is conducive to improving the reader's grasp of the article ideas

2.3Procedure

This paper intends to obey the following process to promote research (Figure 1).

Figure 1. Research flow

  1. Paragraphing needs to be reviewed and restructured.
  • We are really sorry for our careless mistake and thank you for your remainder. We have rewritten the limitations of thestudy. We check out the indentation, punctuation, space, random code and other paragraph problems, and the corresponding places have been modified one by one.
  1. What are the limitations associated with this study?
  • Thanks for your precious suggestion. We have rewritten the limitations of thestudy.The changes to the manuscript are giving in the blue text.

From the perspective of research process, this paper has certain limitations. First of all, the measurement of population system is not complete, and there is a further rich space for the population of the external system of consideration. Secondly, the sample size of this study on the western high-tech zone is slightly insufficient, and the reference value of the conclusions drawn from this study needs further observation. Finally, the contribution coefficients of population and public services to the sustainable development of high-tech zones are not estimated in the article, which worth further discussion in future research.

  1. What are your suggestions for the future research agenda in this field?
  • Thanks for your precious suggestion. We have states the rewritten the suggestions for the future research agendain the last paragraph of Section 4.The changes to the manuscript are giving in the blue text.

From the perspective of future research, this paper has the following expectations. Firstly, referring to the existing research (Bongaarts, 2023), we can bring the external population system's resource and environment factors into the population system's evaluation system under the condition of increasing the availability of data, therefore, the results of the population system development measurement are closer to the reality. Second, broaden the range of sample selection. this research provide us with the value reference (Hua & Hu, 2023). If the conditions permit, we can try to bring more high-tech areas into the research object, so as to get more universal value enlightenment. Finally, an empirical study on the topic of the value contribution of population and public services to regional sustainable development can be conducted. For example, Based on the relevant research (Duan et al., 2023), this paper discusses the causality among the objects of study by VAR model. Based on the relevant research (Ge et al., 2022), the panel model can be used to discuss the effect of population and public services on the level of regional sustainable development. Based on the relevant research (Zhou et al., 2022), we can discuss the population and public services related effects on the level of regional sustainable development with the help of spatial model,.

Reviewer 3 Report

Title: "Regional Population and Public Service under the Framework of Sustainable Development: Evidence from Typical High-Tech Zone in China"

The article presents an examination of the relationship between regional population dynamics and public service provision within the context of sustainable development. The study employs methods such as entropy analysis and the coupling coordination degree model to explore this interplay. The conclusions drawn from the research highlight positive trends in population system evolution and its stable contribution to sustainable development. While the study offers valuable insights into the subject, there could be room for further elaboration on methodology and implications for future research.

Author Response

Response to the Reviewers’ Comments on the PaperSustainability-2576328Entitled“Regional Population and Public Services under the Framework of Sustainable Development: Evidence from Typical High-Tech Zone in China”

August 22, 2023

We would like to thank you for your efforts in reviewing our manuscript and providing many helpful comments and suggestions, which will all prove invaluable in the revision and improvement of our paper, as well as in guiding our research in the future .To facilitate this discussion, we first retype your comments in italic font and then present our responses to the comments.

Response to reviewer #3

1.There could be room for further elaboration on methodology and implications for future research.

  • Thanks for your precious suggestion. As you are concerned, the section of methodology ought to be adjusted. So we have rewritten the section of method.While the section of implication, we have correct some mistakes. The changes to the manuscript are giving in the blue text.
  1. METHOD

2.1. Data

the data in this paper mainly come from the relevant statistical data provided by the Public Security Bureau of high-tech zone and the statistical yearbook of high-tech zone of Chengdu City from 2013 to 2022.

2.2 Method

2.2.1 Entropy method

Sep1, Normalize the initial data.

Treatment of positive indicators:

Treatment of negative indicators:

Step 2, Calculate the weight of item j of Variable i.

Step 3, Calculate the index information entropy.

   Step 4, Calculate the information entropy redundancy.

Step 5, Calculate the indicator weights.

Step 6, The comprehensive score of single indicator.

Choosing the entropy method to construct the evaluation indicator system of population system and public services system. Considering that the strategic core of the development of the high-tech zone of Chengdu city is to respond to the uncertainty of the regional evolution with the strategy of flexible development, this paper uses the entropy method based on the difference-driven principle to obtain the weights of each index, to measure the development level of population system and public services system (Zare, 2023). Its applicability lies in: the processing method of weighting by using the original data is more objective, and the importance of the entropy method to the discrete degree of the data elements is corresponding to the importance of the changing degree of the population condition in the high-tech zone of Chengdu City, the results presented by the use of method are highly explanatory. It should be further explained that the use of the entropy method requires the standardization of data, that is, the conversion of indicators from absolute to relative values, which is intended to solve the problem of homogenization among heterogeneous indicators.

2.2.2Coupling degree model

According to the concept and physical principle of coupling degree model, the following equation can be established.

Establish the coupling degree model of population system and public services system.

Among them, represents the evaluation value of system, C represents the coupling degree, 0≤C≤1. When any evaluation value is 0, the coupling degree is 0; when all evaluation values are 0, the coupling degree is meaningless.

2.2.3Coupling coordination degree model

                  , 0≤D≤1

                   ,

Degree of system coupling and coordination is reflected by the value of D, and the value of T is the comprehensive harmony index of the population system and public services system (0≤T≤1). α and β are the weights of the population system and public services system, respectively.

 The synergy between systems is very important for the change of order parameters from disorder to order, but focusing only on the coupling state may neglect the high-quality operation of the system. In order to explore the coordination between population system and public services system, this paper introduces the criterion of coupling coordination degree (Zhang et al., 2023), the coupling degree and the coupling coordination degree of the high-tech Zone of Chengdu City are calculated based on the positive property of the above-mentioned indexes (Cheng et al., 2023), in order to further explore its element penetration, connectivity level and co-evolution characteristics.

2.3Procedure

This paper intends to obey the following process to promote research (Figure 1).

Figure 1. Research flow

Round 2

Reviewer 1 Report

The authors have tried their best to almost adequately addressed most of my concerns and comments. The manuscript is accepted without re- revision.

Minor editing of English language required

Reviewer 2 Report

Considering the revised version of the manuscript, it is now acceptable for publishing.